# Sensing and Stimulation Applications of Carbon Nanomaterials in Implantable Brain-Computer Interface

**DOI:** 10.3390/ijms24065182

**Published:** 2023-03-08

**Authors:** Jinning Li, Yuhang Cheng, Minling Gu, Zhen Yang, Lisi Zhan, Zhanhong Du

**Affiliations:** 1The Brain Cognition and Brain Disease Institute (BCBDI), Shenzhen Institute of Advanced Technology, Chinese Academy of Sciences, Shenzhen 518055, China; 2Guangdong Provincial Key Laboratory of Brain Connectome and Behavior, Shenzhen Institute of Advanced Technology, Chinese Academy of Sciences, Shenzhen 518055, China; 3CAS Key Laboratory of Brain Connectome and Manipulation, Shenzhen-Hong Kong Institute of Brain Science, Shenzhen Institute of Advanced Technology, Chinese Academy of Sciences, Shenzhen 518055, China; 4Shenzhen Fundamental Research Institutions, Shenzhen 518055, China

**Keywords:** carbon nanomaterials, implantable brain–computer interface, sensing and stimulation

## Abstract

Implantable brain–computer interfaces (BCIs) are crucial tools for translating basic neuroscience concepts into clinical disease diagnosis and therapy. Among the various components of the technological chain that increases the sensing and stimulation functions of implanted BCI, the interface materials play a critical role. Carbon nanomaterials, with their superior electrical, structural, chemical, and biological capabilities, have become increasingly popular in this field. They have contributed significantly to advancing BCIs by improving the sensor signal quality of electrical and chemical signals, enhancing the impedance and stability of stimulating electrodes, and precisely modulating neural function or inhibiting inflammatory responses through drug release. This comprehensive review provides an overview of carbon nanomaterials’ contributions to the field of BCI and discusses their potential applications. The topic is broadened to include the use of such materials in the field of bioelectronic interfaces, as well as the potential challenges that may arise in future implantable BCI research and development. By exploring these issues, this review aims to provide insight into the exciting developments and opportunities that lie ahead in this rapidly evolving field.

## 1. Introduction

Ever since the discovery of electrical impulses in the brain during the 1920s, scientists have been seeking ways to study the brain practically. To advance neuroscience research, improve public health, and increase understanding of the brain, various countries launched research programs such as the United States’ BRAIN Initiative (Brain Research through Advancing Innovative Neurotechnologies) [1] and Neural Engineering Systems Research Program, Europe’s Human Brain Research Program, and Japan’s Brain Mapping Research Program [2]. The United States Army Combat Capability Development Command recently submitted the report “RoboCop 2050: Human-Machine Integration and the Future of Defense” on the future of defense, which identified BCI, visual augmentation, auditory augmentation, and exoskeleton combat suits as transformative technologies closely related to the advancement of neurotechnology. To further develop the BCI field, DARPA launched a series of initiatives, including the Neural Engineering Systems Design (NESD) in 2016, Revolutionizing Prosthetics in 2017, Reconstructing Active Memory (RAM) in 2018, Next-Generation Non-Invasive Neurotechnology (NGNIT) in 2018, Next-Generation Nonsurgical Neurotechnology(N3) in 2018, and Bridging the Gap Plus (BG+) projects in 2019. These initiatives have allowed for the advancement of BCI from invasive, single-task brain interfaces to sophisticated, multitask spinal interfaces. Notably, the recent growth of BCI manufacturing firms such as Neuralink has increased public awareness of this technology. As the development of BCI has raised the bar for neural information flow, the study of implantable neural electrode devices and materials has emerged as a frontier field of brain science research [3,4,5,6,7].

BCIs have enabled high-throughput information interchange between the human brain and external mechanical devices, leading to functional integration. For over 50 years, neurophysiologists have employed various electrode materials implanted in the brain to study brain activity. However, prior to the widespread use of carbon nanomaterials, implantable electrode materials were predominantly metallic, including tungsten, gold, platinum, copper, stainless steel, silicon, with complexes such as Michigan, Utah, and microfilament electrodes [8,9,10,11]. The biocompatibility and physical structural properties of these materials are limited to the material itself. The introduction of fullerene by Smalley et al. in 1985 at Rice University and Sussex University marked the first member of the class of carbon nanomaterials and initiated a new wave of research on their application [12]. The subsequent discovery of carbon nanotubes and graphene in 1991 and 2004, respectively, stirred the scientific community [13,14], as did carbon quantum dots and nanodiamonds. These low-dimensional carbon nanostructures possess unique optical, electrical, magnetic, and chemical properties that differentiate them from basic carbon materials like diamond and graphite, opening up new avenues for research on carbon material interfaces for implanted nerve electrodes.

Carbon nanomaterials have emerged as promising candidates for biomedical applications in the past decade, owing to their unique physical and chemical properties. Unlike other nanoparticle materials, carbon is biocompatible and non-toxic, which makes it an ideal material for implantable neural interfaces [15]. The similar size of nanoparticles and certain essential proteins makes them suitable for in vivo applications [16]. The small size of nanocapsules and nanocarriers makes them useful for loading and delivering drugs and genes across the blood-brain barrier to specific sites in the brain [17]. Carbon nanomaterials offer superior charge injection capabilities and high conductivity, enabling high-throughput electrode interfaces that can enhance signal recording quality and stimulation efficiency. In addition, the optical properties and chemical stability of carbon nanomaterials, along with their large surface area, make them ideal for surface charge modification and the incorporation of fluorescent tags, cell-specific targeting molecules, and disease-specific targeting molecules [18,19]. The lightweight, porous, flexible, conductive, and stable nature of carbon nanoframes makes them a useful tool for neural tissue engineering, allowing for the enhancement of electrode flexibility [20]. Figure 1 summarizes the various applications of carbon nanomaterials in implantable BCI, highlighting the material’s superiority and potential for advancing neuroscience research and health applications via BCI.

## 2. Properties and Sample Applications of Different Carbon Nanomaterial

### 2.1. Zero-Dimensional Carbon Materials: Fullerene and Nanodiamond

Nanoparticles, quantum dots, and nanoclusters, which possess dimensions on the order of nanometers in all three directions, are considered zero-dimensional nanomaterials. Carbon nanomaterials such as nanodiamond and nano-fullerene C60 are common examples of zero-dimensional carbon nanomaterials that are utilized in implantable brain interfaces.

Nanodiamonds are a type of carbon-based crystal with a diamond-like structure and minimal cytotoxicity [21]. During production, Type Ib (dispersed state) nanodiamonds are annealed at high temperatures to create a nitrogen-vacancy (NV) color center. This defect center has strong absorption and emission at 560 nm and is located deep within the nanodiamond core, making it unaffected by surface chemistry [22]. The distinct fluorescence of negatively charged NV color centers can be selectively controlled by spin manipulation, making it useful in ultrasensitive biosensing applications [23]. The fluorescence change of NV color centers can be modulated by lasers and microwaves to achieve ultrasensitive sensing [24,25,26]. Moreover, nanodiamond surfaces can be modified to enable biomolecule binding. Thin films can be selectively biomodified and adsorbed to integrate DNA and other biomaterials with microelectronics to create bioelectronic sensing systems [27]. Nanodiamonds can also be used for cellular tracing in vivo for cancer cell and stem cell division and differentiation [28], as a coating material to promote the formation of functional neuronal networks [29,30], and as a fluorescent nanodiamond-assisted cellular in vivo imaging technology, among other applications [30].

Fullerenes are a zero-dimensional class of conjugated spherical molecules composed of sp2 hybridized carbon atoms that exhibit unique topological structure and photoelectrochemical properties compared to other carbon nanomaterials, including strong photoelectric and photothermal conversion effects, long-lived triplet states, and high visible-light absorption ability. Recent years have seen interesting applications of C60-based materials in analytical sensing. The properties of fullerenes are optimized by synthesizing C60 derivatives and non-covalent modifications and then employing them for the electrochemical and photoelectrochemical sensing of biomolecules. Chaniotakis et al. first described a fullerene-mediated electrochemical biosensor [31] for glucose. In this biosensor, C60 is adsorbed in a porous carbon electrode for electron transport. C60 serves as an efficient electron acceptor [32] and exhibits rich electrochemical behavior owing to its high number of conjugated double bonds and low-vacancy LUMO orbitals, endowing it with a strong electron acceptor ability. Furthermore, the unique cage structure of fullerenes enables the insertion of other small molecules into their cavities, leading to significant modifications in the physical and chemical properties of fullerene molecules in applications such as tumor imaging [33,34,35] and drug loading [36].

### 2.2. One-Dimensional Carbon Materials: Carbon Nanotubes

Carbon nanotubes (CNTs) are a remarkable one-dimensional hollow tubular nanomaterial with an ultrahigh aspect ratio. They possess excellent electrical conductivity and flexibility, as well as a unique high aspect ratio and surface area, making them an ideal candidate for hosting drugs or biospecific molecules. In addition, CNTs exhibit exceptional drug loading capacity via covalent or non-covalent interactions. These characteristics have propelled CNTs as promising materials for various biomedical applications. In particular, their distinct electrical properties enhance biocompatibility investigations for brain applications. Furthermore, the needle-like shape of CNTs allows them to penetrate the cytoplasmic membrane, enabling the delivery of drugs across the membrane. These attributes make CNTs an exciting platform for drug delivery and medical applications.

Carbon nanotubes possess unique physicochemical properties that make them highly attractive for use in various neurological applications. One key advantage of these nanomaterials is their low impedance, high charge transfer capabilities, high sensitivity, and easy modifiability, which make them ideal candidates for use as electrode materials [37,38,39]. For instance, Gross et al. demonstrated that carbon nanotube coatings can reduce electrode impedance, increase charge transfer, and enhance the recording and stimulation properties of neural electrodes, enabling the sensitive and selective detection of in vivo diseases [40]. Lee et al. developed carbon nanotubes with a unique structure that can be utilized for a wide range of applications, including artificial synapses with proprioceptive feedback that ensure a steady transmission of nerve impulses to the muscles [41]. However, primitive carbon nanotubes are not completely soluble in all solvents, which can cause specific health issues. To address this, researchers are investigating their biological features in terms of toxicity [42]. The discovery of efficient techniques for chemically modifying carbon nanotubes has accelerated the development of soluble forms for a variety of biological applications, including drug administration [43,44,45,46]. To be effective in delivering therapeutic agents, a carrier must be able to efficiently penetrate cells. Functionalized carbon nanotubes (f-CNTs) have the ability to traverse cell membranes and localize to specific cellular compartments when treated with fluorescein isothiocyanate (FITC) or fluorescent peptides [47].

### 2.3. Two-Dimensional Materials: Graphene and MXene

Recently, there has been a growing interest in two-dimensional (2D) functional materials, specifically graphene. Graphene is a fundamental building block for various carbon materials, as it can be manipulated into zero-dimensional fullerenes and one-dimensional carbon nanotubes, and stacked into three-dimensional graphite [48]. Graphene consists of carbon atoms connected by sp2 hybrid orbitals, forming a carbon–carbon bond length of 0.142 nm [49]. The remaining electron in the carbon atom’s p orbital produces the enormous π-bond, which contributes to the structural stability of graphene. The interaction between electrons in graphene and the periodic potential of its honeycomb lattice produces new quasiparticles known as massless Dirac fermions, which exhibit unique properties such as the anomalous quantum Hall effect [50,51]. Due to its exceptional electron mobility and electrocatalytic activity, graphene has emerged as a promising material for biosensing electrodes, making it increasingly relevant for a broad range of neurological applications [52,53,54].

MXenes refer to a group of two-dimensional (2D) metal carbides and nitrides that possess a honeycomb-like structure, consisting of numerous layers of transition metal (M) atoms [55,56,57]. These materials are produced by selectively etching A-layer atoms from the MAX phase, resulting in loosely stacked MX layers, which are commonly known as “MXene” and can be further separated into individual monolayer sheets [55]. It is predicted that metallic MXene monolayers possess a high electron density at the Fermi energy level [58,59]. MXenes exhibit robust electrical conductivity, feature abundant functional groups like hydroxyl, fluorine, and oxygen, and exhibit varying surface characteristics, which make them an excellent choice for bio-electronic interfaces.

## 3. Applications of Carbon Nanomaterials in the Field of Implantable BCI

### 3.1. Recording and Stimulation Performance of Implantable BCI Electrodes

Efficient regulation of cellular electrical activity in specific neuronal target groups is critical for fundamental neuroscience research and associated BCI applications. Carbon nanoparticles have recently emerged as a focus of intense study due to their unique electrical and physicomechanical properties, making them a promising alternative to traditional metal or silicon electrode materials in neural interface systems. In 2008, Keefer et al. demonstrated the potential of carbon nanotube-coated tungsten and stainless steel neural recording electrodes to enhance charge transfer and lower electrode impedance by a factor of 23, thereby significantly improving neuron recording and electrical stimulation in cultures, rats, and monkeys (Figure 2a–c) [40]. The modified electrodes exhibited increased durability, decreased resistance, reduced susceptibility to noise, and an enhanced ability to activate neurons. The properties of individual carbon nanotubes and the electrode-surface deposition process influence the functional qualities of the neural interface. While traditional carbon nanotube films are brittle and generated via solvent evaporation [60,61,62,63,64], electrochemical deposition [40], and chemical vapor deposition [65,66,67,68], composites of carbon nanotube multilayer and polyelectrolytes fabricated through layer-by-layer (LBL) assembly can significantly improve plating durability [69]. In 2009, Jan et al. at the University of Michigan evaluated composite multiwall carbon nanotube-polyelectrolyte electrodes with better interfacial performance than conventional iridium oxide (IrOx) and poly(3,4-ethylene dioxythiophene) (PEDOT) electrodes (Figure 2d–f), effectively reducing the voltage difference during nerve tissue stimulation, demonstrating higher electrochemical stability and overcoming structural challenges such as electrode coating cracking and delamination, making it a promising new material for neural interfaces [70].

Subsequent to the research on carbon nanotubes and their application in neural interface systems, polypyrrole/carbon nanotube composites [71], pure carbon nanotube (CNT) doped poly(3,4-ethylene dioxythiophene) (PEDOT) [72], and various other composites have been utilized to improve conventional microelectrodes. These include Pt microelectrodes [71,72], conical silicon (cs-si) [73], and microfilament microelectrode arrays [74], as well as carbon nanotube-modified electrodes and complex coatings. The modified electrodes exhibit improved charge storage capacity (CSC) and reduced electrochemical impedance, which can significantly enhance both recording and stimulation performance.

Graphene coatings have become popular in the development of neural electrodes [75,76,77]. Graphene-based electrodes have been found to provide superior performance compared to conventional electroretinogram (ERG) electrodes, with higher signal amplitudes and better biocompatibility [77]. Additionally, graphene is being studied for use in implanted brain interfaces. Liu et al. developed high-performance graphene/Ag electrodes by modifying traditional metal electrodes with graphene to form a van der Waals heterostructure at the interface. This modification significantly improved the electrochemical performance of the graphene/Ag electrodes, with enhanced electron transfer and an ultra-high specific surface area, which led to improved detection of LFP signals with greater oscillation amplitudes than conventional Ag electrodes. This increased the SNR by a factor of 2.4 and improved neural signal acquisition sensitivity [78] (Figure 2g–i). In 2020, Xiao et al. developed reduced graphene oxide (rGO) and platinum (Pt) black coatings that were electro-deposited on planar Pt microelectrodes, resulting in a >60-fold decrease in impedance, which permitted the long-term detection of neural spike potentials in epileptic mice [79]. Guimerà-Brunet et al. developed a graphene-based field-effect tube array that can capture sub-low brain activity across a vast area, unlocking information below 0.1 Hz [53]. Wang and colleagues developed graphene-fiber-based microelectrode arrays with a thin platinum covering, resulting in low impedance and a multi-null structure of GF, which lead to unparalleled charge injection capabilities, and the thin platinum layer effectively conveyed the gathered signals down the microelectrode, resulting in a powerful synergistic impact between the two components. In vivo research has shown that microelectrodes implanted in the rat’s cerebral cortex can detect neural activity in regions as small as individual neurons with a signal-to-noise ratio of micro 9.2 dB [60]. Nicolas A Alba et al. employed electrochemical techniques to characterize carbon nanotube (MWCNT) and dexamethasone (Dex)-doped poly(3,4-ethylenedioxythiophene) (PEDOT) coatings, which showed promise for improving chronic neural electrode performance [80].

Nanocrystalline diamond (NCD) is an emerging carbon-based material for developing high-resolution implanted brain connections. Nitrogen-doped ultra-nanocrystalline diamond (N-UNCD) has been proposed as an electrode material for local stimulation in high-density 3D column electrode arrays of high-sensitivity retinal prostheses, owing to its high chemical and biochemical inertness, biocompatibility, and long-term stability. In addition, Hejazi et al. have selectively deposited N-UNCD on carbon-fiber microelectrodes for neurostimulation, high-quality single-unit neural recordings, and neurotransmitter detection [81] (Figure 2j–l). However, electrode implantation can cause scar development and inflammatory tissue damage, which reduce the neuron density at the electrode-tissue interface, form a high impedance layer, and lower the electrode performance. Therefore, many research groups have explored strategies to overcome these challenges. For instance, the on-demand delivery of anti-inflammatory drugs and neurotrophic factors has shown considerable promise in maintaining a stable chronic neural interface. These approaches offer potential solutions to the technological problems associated with electrode implantation [82,83].

**Figure 2 ijms-24-05182-f002:**
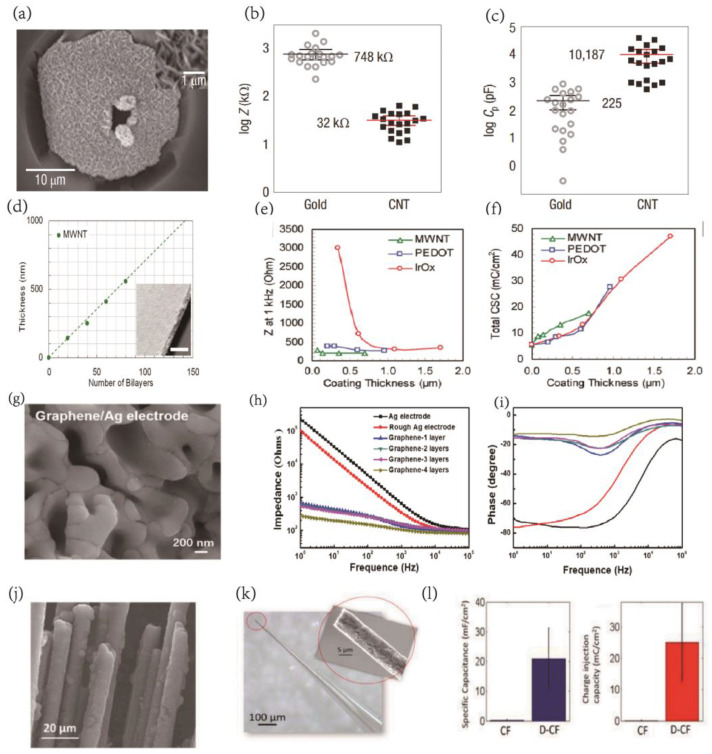
(**a**) SEM image of CNT−coated MEA electrode (∼20 µm diameter). The crater was formed by ablating the overlying dielectric layer to access the indium−tin oxide conductor. Inset: high magnification reveals the porous character of the CNT coating. (**b**,**c**) The CNT coating led to a 23−fold decrease in impedance and a 45−fold increase in charge transfer. Reprinted with permission from [40] Copyright © 2023, Springer Nature. (**d**) MWNT coating thickness as a function of deposition parameters. (**e**,**f**) Comparison of electrochemical properties on a per unit film thickness basis. The coating−thickness normalized measurements were obtained from electrochemical impedance spectroscopy and cyclic voltammetry. Reprinted with permission from [70] Copyright © 2023 American Chemical Society. (**g**) The graphene/Ag electrodes in different scale magnifications. (**h**,**i**) Bode plot of (**h**) impedance and (**i**) phase of electrodes as a function of the frequency. (**j**) diamond−coated CFs. Reprinted with permission from [78] Copyright © 2023 American Chemical Society. (**k**) Optical image of a single diamondecoated CF electrode encapsulated in a glass pipette with the SEM image of the electrode tip shown in (**j**) inset. (**l**) Specific capacitance increased from 0.263 ± 0.168 mF/cm^2^ on CF to 20.90 ± 10.30 mF/cm^2^ on D−CF electrodes; CIC of D−CF was 238 times larger than that of CF, which increased from 0.105 ± 0.067 mC/cm^2^ to 25.08 ± 12.37 mC/cm^2^. Reprinted with permission from [81] Copyright © 2023 Elsevier.

### 3.2. Flexible Electrode Arrays

Recording and stimulating intracranial and peripheral nerves are crucial for studying fundamental neurological processes and treating neurological disorders. However, traditional electrode implantation materials such as metal and silicon often cause injury, inflammation, and electrode corrosion, leading to impaired long-term durability of the implants. Moreover, implantable microelectrodes such as depth electrodes only sample limited data in specific anatomical regions, making large-area brain mapping challenging. To overcome these issues, carbon nanoparticles’ structural flexibility provides an excellent mechanical match to the surrounding tissue, which can significantly minimize neural tissue damage and inflammatory reactions. Carbon nanomaterials are replacing conventional conductive materials such as silicon and metal for implantation into deep brain tissue to develop innovative neural interfaces. Flexible brain electrodes made of carbon nanoparticles with coated structures are commonly employed for this purpose. Optically transparent and flexible graphene electrodes are particularly effective in electroretinogram conformal entire cornea recordings, providing a larger signal amplitude than standard ERG electrodes (Figure 3a–d). Furthermore, Vitale et al. employed the strong hydrophilicity of Ti_3_C_2_ MXene to create conductive inks for developing microelectrode arrays for in vivo neuronal recordings [84].

Duygu Kuzum’s group utilized direct laser pyrolysis to fabricate porous graphene dots on polyimide films [84], which were then used to construct flexible microelectrode arrays for brain signal stimulation and sensing [85,86]. These microelectrode arrays were designed to meet the high spatiotemporal resolution requirements necessary for identifying cortical dynamics. The low-impedance porous graphene electrodes are highly sensitive to cortical patterns, resulting in high signal-to-noise ratios. The flexibility of the electrodes enables their application to the brain and joints, and in vivo testing in mice has confirmed their effectiveness in recording different brain activity patterns, such as standing waves, moving plane waves, and spirals (Figure 3e–g). Laser pyrolysis allows for the direct growth of polyimide on the substrate, which eliminates delamination problems associated with coating. The impedance value of the porous graphene electrode is approximately two orders of magnitude lower than that of a gold electrode of the same size. Chemically doped graphene produced by nitric acid treatment further decreases the impedance value while increasing the Charge Injection Limit (CIL) from 2 mC/cm^2^ to 3.1 mC/cm^2^.

Soft nerve electrodes made of graphene and carbon nanotube (CNT) fibers offer a promising solution for sustained, chronic in vivo action potential recordings with a far lower inflammatory response. The electrodes’ porosity microstructure results in low electrochemical impedance, allowing for a much smaller electrode size. Additionally, the low Young’s modulus of the fibers results in exceptionally soft electrodes. Buschbeck et al. demonstrated the effectiveness of functionalized carbon nanofibers as implanted electrodes for recording signals in the optic lobes of insects such as cockroaches, with longer in vivo recordings, less scar tissue formation, and reduced cytotoxicity compared to conventional metal electrodes [87]. However, Pt films, which are less flexible and stretchable than CNT films, may suffer from damage and delamination. While extremely flexible micro-wire electrodes may target deeper brain areas with minimal tissue injury, CNT fibers can be spun into tiny, flexible fiber microwires (10–100 µm in diameter) through dry or wet spinning, resulting in low impedance and high CIC electrical properties [88]. Graphene oxide fibers can also be wet-spun and annealed to create liquid crystal graphene oxide (LCGO) fiber carbon nanomaterials, which have outstanding structural features suitable for flexible electrode implantation with 3D architectures for interconnecting with nerve fiber structures [89] (Figure 3h).

Transparent nerve electrode arrays made of graphene or carbon nanotubes (CNT) can be used in conjunction with various neural interface modalities, including electrophysiological measurements, optical imaging/stimulation [54], and magnetic resonance imaging (MRI) [90] (Figure 3i). The use of flexible electrodes presents a challenge for implantation, and researchers such as Apollo et al. have proposed various solutions that rely on chemical, mechanical, fluidic, or magnetic modalities [91]. The spatial and temporal resolution advantages of different approaches can be leveraged, yielding new insights into how brain circuits process information. By combining these methods, it may be possible to achieve targeted and minimally invasive implantation of these electrodes in specific brain areas without causing damage or bending.

**Figure 3 ijms-24-05182-f003:**
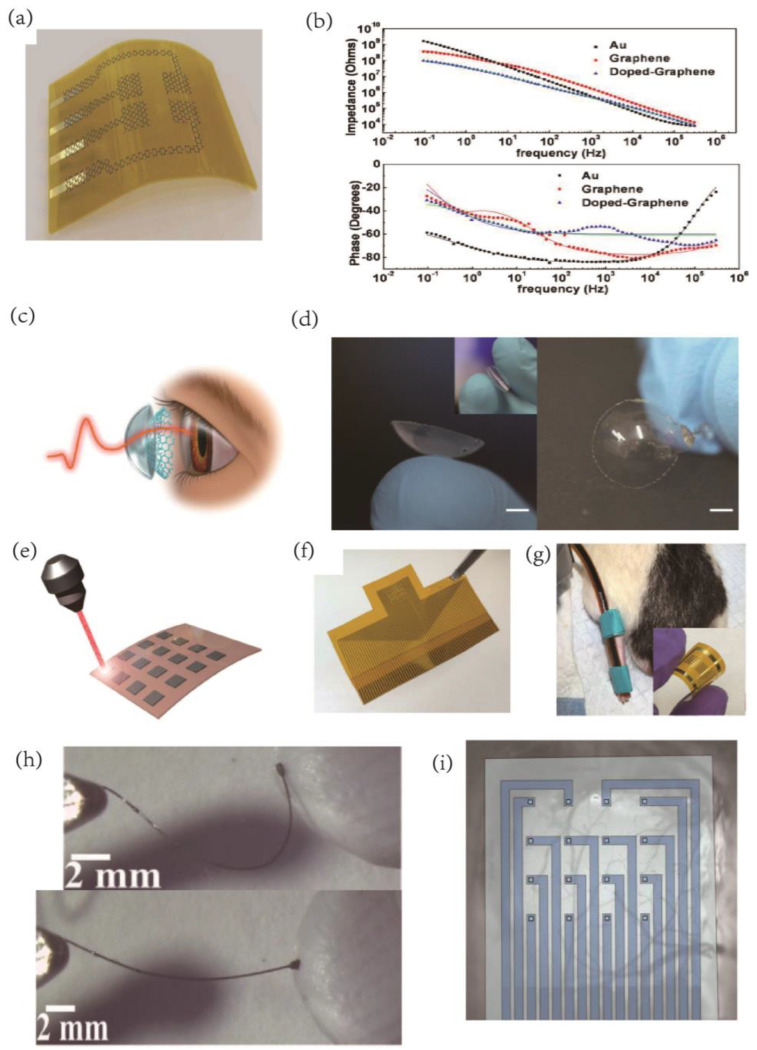
(**a**) Schematic illustration of a flexible graphene neural electrode array. (**b**) EIS results for 50 × 50 µm2 Au, G, and doped graphene samples. Measurement results are shown with symbols, and regression results are shown with solid lines for impedance magnitude (**top** figure) and phase (**bottom** figure) plots. The impedance magnitude (**top** figure) significantly decreased with the doping of graphene, more prominently for frequencies lower than 1 kHz. Reprinted with permission from [75] Copyright © 2023, Springer Nature. (**c**) Schematic drawing of ERG recording with the GRACE device. (**d**) Photographs of a GRACE device made from G−quartz. Scale bar, 3 mm. The image in the inset demonstrates the high softness of the GRACE device. Reprinted with permission from [77] Copyright © 2023, Springer Nature. (**e**) Porous graphene is fabricated with laser pyrolysis. (**f**) A picture of the 64−channel porous graphene electrodes array. The scale bar: 1 cm. (**g**) A resistive flex sensor spanning the knee joint. The inset illustrates the flexibility of the 16−electrode array as fabricated. Reprinted with permission from [86] Copyright © 2023, Springer Nature. (**h**) LCGO electrode pressed into clay and released to demonstrate flexibility and elastic deformation. Reprinted with permission from [89] Copyright © 2023, Wiley. (**i**) The device is >90% transparent across the visible to near−infrared spectrum. Reprinted with permission from [54] Copyright © 2023, Springer Nature.

### 3.3. Electrochemical Sensing of Biomolecules

In recent years, carbon nanomaterial biosensor research has seen a significant surge in interest. With BCI increasingly being used as diagnostic tools for disease detection, supplementary cellular information and high temporal resolution electrical recordings are necessary to create multiple biosensing channels. For these applications, sensing materials with high selectivity, sensitivity, rapid response, and extended shelf life are essential. Carbon nanoparticles are among the most promising materials for biosensing applications, owing to their unique characteristics such as solid electrical conductivity, chemical stability, and large active surface area. Electrochemical sensors for biomolecules such as dopamine, ascorbic acid, and serotonin, hydrogen peroxide, proteins such as biomarkers and DNA, as well as non-electrically active substances like glucose, alcohols, and proteins, and nitric oxide, can be created using a variety of strategies involving carbon nanomaterials.

Neurotransmitters (NTs) are crucial chemical messengers for neurotransmission [92] and have a significant role in BCI. Therefore, neurotransmitter sensing has gained considerable attention in neuroscience research. Changes in the concentration of neurotransmitters in the central nervous system have been associated with various physical and mental illnesses, including Parkinson’s disease, Alzheimer’s disease, depression, schizophrenia, and drug addiction [93,94,95,96,97]. Since the 1970s, electrochemical techniques have been developed to measure neurotransmitters, including galvanometric and voltammetric methods. Among these, fast scanning cyclic voltammetry (FSCV) with carbon-fiber microelectrodes has emerged as a practical approach for detecting electrochemically active neurotransmitters (NT) or neuromodulators (NM) [98] (Figure 4a). Carbon material electrodes, which facilitate electron transport and surface oxide adsorption due to electrostatic interactions, are ideally suited for detecting neurotransmitters [99,100], especially with the use of carbon nanomaterials. Surface biomodification of nanoparticles can be achieved through thiol–metal interactions, avidin–biotin interactions, pi-stacking interactions, and the NHS–EDC carbodiimide reaction [101]. Nanomaterials can interact with biomolecules such as proteins, enzymes, and DNA to create nanobiosensors with high selectivity, sensitivity, and extended shelf life for biochemical and physical response sensor design.

Yang et al. have reported the development of carbon nanotube microelectrodes on niobium substrates for dopamine detection, exhibiting a detection limit of 111 nM, which is nearly twice that of bare carbon-fiber electrodes [102]. In another study, Bao et al. fabricated flexible neuro-transmitter sensors based on graphene-elastomer composites to monitor the kinetics of monoamine neurotransmitters in vivo in living animal brains and intestines [52] (Figure 4b). To address the limitations of stiff and brittle carbon-fiber electrodes and their restricted tunability, MXene has been demonstrated to be easier to micromachine than graphene, and it has been used to create MXene-based field effect transistors for dopamine detection and the rapid detection of primary neuronal action potentials [103] (Figure 4c). The use of two-photon nanolithography to produce freestanding carbon microelectrodes for implantable neurochemical sensing has allowed for the mass production of carbon nanomaterial electrodes. These electrodes have been shown to be highly accurate in detecting other well-known neurochemicals such as ascorbic acid, serotonin, epinephrine, and norepinephrine, as demonstrated in studies by Venton et al. [104,105]. Moreover, carbon nanomaterial electrodes can be employed for detecting various other biochemical indicators of the brain, such as glucose, 1-lactic acid, ascorbic acid (AA), nitric oxide, and inflammatory factors, among others. These critical physiological parameters are essential for brain disease diagnosis, and carbon nanomaterial electrodes can fulfill the corresponding detection tasks with high sensitivity and selectivity [106,107,108].

In vivo detection of ascorbic acid (AA) is essential for understanding brain chemistry, but it poses a challenge due to interference from other electroactive compounds. To tackle this issue, Zhang et al. [109] developed carbon nanotubes modified with multi-walled carbon nanotubes (MWNT), which enhanced ascorbic acid oxidation for more accurate detection. By employing a novel electrochemical technique, they selectively oxidized AA and prevented interference from other electroactive compounds in the brain, such as 3,4-dihydroxyphenylacetic acid (DOPAC) and uric acid (UA). Traditional electrodes, such as Au, Pt, or glassy carbon, have overlapping oxidation potentials for AA and dopamine (DA), making their independent detection challenging. Gao et al. [110] addressed this issue with a graphene oxide (GO)-modified glassy carbon (GCE) electrode for DA detection in the presence of AA analyte interference. Halima et al. [111] used chitosan (CS) and graphene sheets (GS) to create electrochemical sensors for the detection of DA and UA. The amidation functionalization of CS-GS was found to improve electrocatalytic performance in oxidation potential and peak current. The CS-GS matrix functioned at lower detection limits (0.14 and 0.17 M) with extensive linear ranges (1–700 and 1–800 M) and high sensitivity (2.5 and 2.0 AM^−1^ cm^−2^) for DA and UA, respectively. In complex environments with several combined substances, accurately differentiating between compounds of similar types with dependable temporal and spatial resolution remains the most challenging task for biochemical sensors used in vivo.

**Figure 4 ijms-24-05182-f004:**
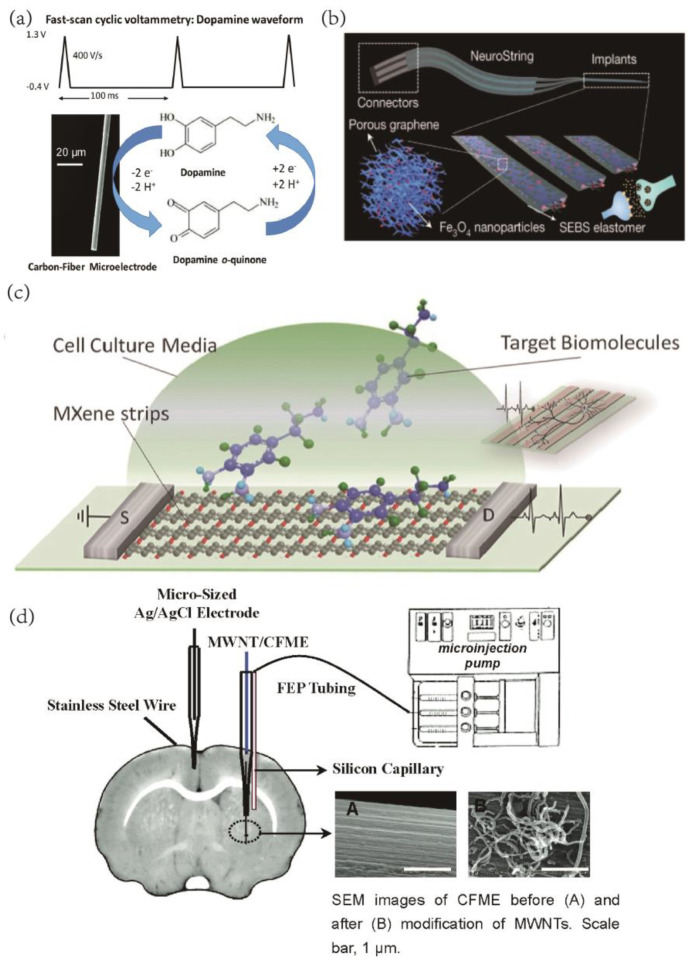
(**a**) FSCV waveform for dopamine. Reprinted with permission from [98] Copyright © 2023 The Royal Society of Chemistry. (**b**) Schematic of the soft implant for sensing neurotransmitters in the brain, and 3D schematic showing the composite materials made by confining nanoscale graphene/iron oxide nanoparticle networks in an elastomer (SEBS) to construct a soft, sensitive, and selective neurochemical sensor. Reprinted with permission from [52] Copyright © 2023, Springer Nature. (**c**) Schematic of a biosensing device based on MXene field−effect transistors. Reprinted with permission from [103] Copyright © 2023, Wiley. (**d**) Schematic Illustration of Experimental Setup for Vivo Voltammetric Measurement of Striatum AAaa with the MWNT−modified electrode and exogenous infusion of standard AA and AAox into rat striatum through a silicon capillary and FEP tubing pumped with a microinjection pump. Reprinted with permission from [109] Copyright © 2023, American Chemical Society.

### 3.4. Drug Delivery

In addition to their ability to directly regulate brain tissue via electrophysiological properties, nanomaterials that have been modified and processed to transport medicines for targeted delivery to specific locations represent another promising avenue for neuromodulation. Carbon nanotubes, a type of carbon nanomaterial, possess a unique inner lumen structure that makes them highly suitable for drug release. For instance, they can be used as nano-reservoirs to contain controlled pharmaceuticals for targeted administration through non-covalent functionalization. In one study, electrical stimulation was used to control the re-release of anti-inflammatory dexamethasone [112] (Figure 5a). Carbon nanotube-modified electrodes also hold potential for highly precise pharmacological release, which could be used to influence neuronal activity with excellent temporal and spatial accuracy. For instance, one study used electrodes to release DNQX, an AMPA receptor antagonist, in a rat brain with precise control and was able to alter cortical sensory coding in rats [113] (Figure 5b). This research could lead to the development of a closed-loop BCI with exceptional temporal precision in disease therapy, surpassing traditional injection-based methods.

Controlled drug release is a promising area of research for the treatment of neurological disorders, and various materials have been investigated for their potential in this field. In addition to graphene, which is a vital material in the development of bioelectronic interfaces, carbon nanomaterials, such as carbon nanotubes have been studied for their ability to serve as nano-reservoirs for targeted drug delivery. For example, Zhu et al. developed a graphene/polypyrrole composite electrode (GN-PPy-FL) to release sodium fluorescein (FL) via voltage [114]. Similarly, He et al. used a reduced graphene oxide (rGO)-DOX modified flexible electrode to electrophoretically deposit adriamycin (DOX) onto the rGO film and release it with a positive potential pulse [115] (Figure 5c). Du et al. developed a novel dual-layer conductive polymer/acid functionalized carbon nanotube (fCNT) microelectrode coating to better facilitate the loading and controlled delivery of the neurochemical 6,7-dinitroquinoxaline-2,3-dione (DNQX) [113]. In another study, electrostatically spun PLGA and graphene oxide composite were found to release IGF-1, promoting neural stem cell development, a promising approach for in vivo implantation to aid in brain repair [116] (Figure 5d). The use of electro-responsive drug release platforms, such as conductive polymer polypyrrole (PPy) nanostructures doped with graphene-mesoporous silica nanohybrid (GSN) nanostructures, can provide on-demand controlled drug delivery with spatial and temporal control. Additionally, electrochemical stimulation has been shown to promote peripheral nerve regeneration and to protect cells from toxicity associated with Alzheimer’s disease [117]. These studies offer insight into the potential of nanomaterials for targeted drug delivery and neural regeneration, and pave the way for further research on the development of implantable BCI.

In the realm of drug delivery, 0-dimensional carbon nanoparticles have emerged as a promising approach to treating illnesses non-invasively by transporting drugs or neurotransmitters across the blood-brain barrier (BBB). To achieve this, functionalized carbon dots have been employed to create various nanoparticle materials for drug transport over the BBB. For instance, a recent study utilized transferrin to enable carbon nanodots to cross the BBB and reach the brain, and this study also evaluated the effectiveness of this modification through zebrafish delivery experiments [118]. Although no carbon dots are currently available as in vivo microelectrode drug release materials, their intrinsic conductive properties make them an attractive option for modification on the surface of BCI electrodes. In addition, nanofluidic drug-release platforms composed of fullerenes have been shown to enhance the temporal precision of drug delivery [119] (Figure 5e–g). Given the exceptional qualities of fullerenes and their drug release properties, this technology may have important applications in the nervous system.

**Figure 5 ijms-24-05182-f005:**
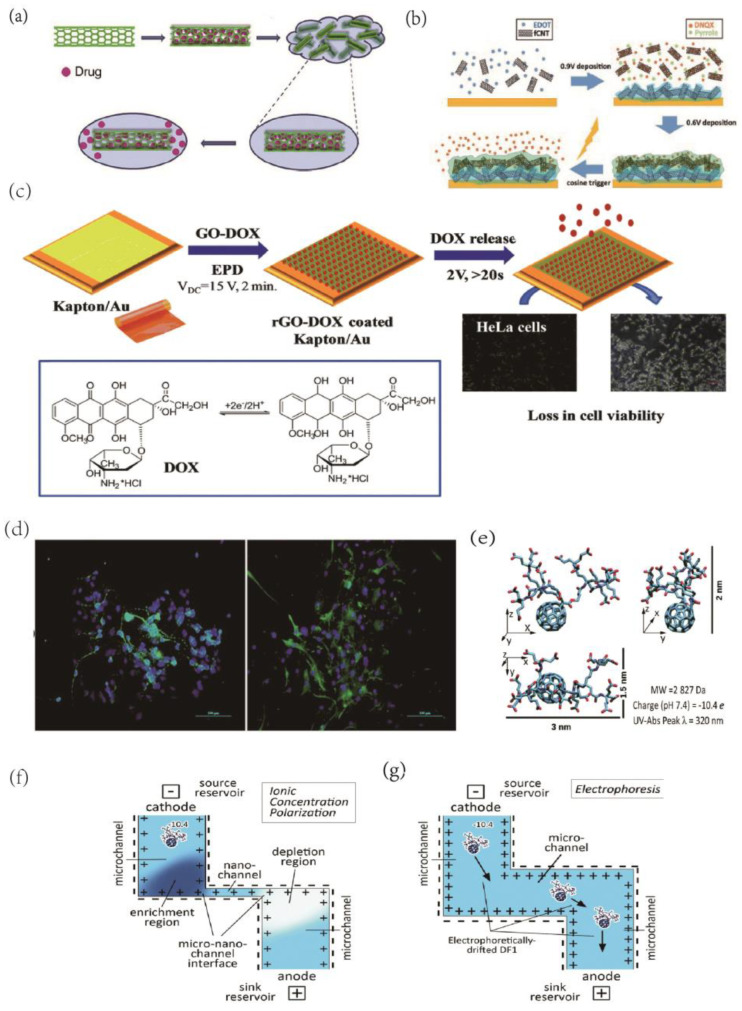
(**a**)Schematic of CNT nano reservoirs’ drug loading and release process. Reprinted with permission from [112] Copyright © 2023, Elsevier. (**b**) Illustration of the synthesis of dual−layer PEDOT/fCNT−PPy/fCNT/DNQX film and controlled release of DNQX from the film. Reprinted with permission from [113] Copyright © 2023, Wiley. (**c**) Schematic representation of the formation of Au/Kapton flexible electrodes coated with doxorubicin−loaded reduced graphene oxide (rGO-DOX) upon application of a VDC = 15 V for 2 min followed by the electrochemically triggered desorption of the drug and the in vivo test of DOX activity on HeLa cells. Reprinted with permission from [115] Copyright © 2023, The Royal Society of Chemistry. (**d**) Immunofluorescence staining was performed in NSCs after seven days of differentiation. The markers are green, while the cell nuclei, counterstained with DAPI, are blue. All scale bar lengths are 100 μm. Reprinted with permission from [116] Copyright © 2023, The Royal Society of Chemistry. (**e**) The structure and properties of DF−1. Schematic of nanochannel Delivery System (nDS) membrane under (**f**) ionic concentration polarization (ICP) effect in a slit nanochannel and (**g**) electrophoretic effect in a microchannel. Reprinted with permission from [119] Copyright © 2023, The Royal Society of Chemistry.

### 3.5. Neural Tissue Engineering

Neuronal healing has posed a significant challenge due to the complex structure and function of the nervous system. In the field of neural tissue engineering, strategies using autophagic and heterophonic cells have been employed to promote neurite development and nerve regeneration. However, limitations associated with donor site morbidity and the scarcity of donor tissue have motivated researchers to explore alternative sources. Tissue engineering is a promising therapeutic approach that combines basic principles of cell biology, such as cell adhesion, differentiation, and value addition, with bioengineering. By utilizing bioengineered structures such as scaffolds, lamellas, and nanoparticles, neural tissue engineering creates an environment that facilitates nerve cell repair and regeneration.

Carbon nanotubes (CNTs) are among the most commonly used carbon nanomaterials in brain tissue engineering due to their remarkable electroactive properties, which make them ideal for interaction with electroactive tissues. Several studies have demonstrated the ability of carbon nanotube substrates to support neuronal survival and enhance neuronal development [63,120,121] (Figure 6a). Mazzatenta et al. [64] developed an in vitro model for neuron/SWNT coupling by utilizing the biocompatibility and adhesion properties of single-walled carbon nanotubes (SWNTs), which allowed rat hippocampal neurons to grow on purified SWNT films and triggered membrane depolarization responses upon electrical stimulation delivered through the SWNT layer.

In tissue integration, carbon nanotubes have remarkable significance in both the 2D structure and the 3D structure–substrate interface [122]. Bosi et al. created an elastic scaffold for 3D primary neuron development and demonstrated that the 3D network topology was more effective [123] (Figure 6b). Usmani et al. proposed a self-supporting framework made of multi-walled carbon nanotubes organized in a three-dimensional (3D) lattice to successfully connect morphologically separated neural stages in vitro, leading to the spontaneous regeneration of neuronal bundles that were shaped into a dense random network [124] (Figure 6c). This 3D scaffold morphology has potential applications in the development of organotypic spinal cord growth and functional reconnection, as well as in the development of peripheral nerve prosthetic devices.

Graphene, the lightest and stiffest atomic layer structure known to humans, is also the material with the highest thermal conductivity at ambient temperatures. These exceptional properties make graphene an ideal candidate for constructing scaffolds for cell growth and differentiation in brain tissue engineering. To this end, Martin et al. employed various methods to create hybrid hydrogels consisting of polyacrylamide and graphene, demonstrating that the inclusion of graphene (GR) enhanced the neuronal biocompatibility of the 3D scaffolds [125].

Carbon nanoparticles have emerged as a promising alternative in brain tissue engineering due to their exceptional performance and ease of production. Lee and colleagues developed a bio-ink called MWCNT-PEGDA and employed stereolithography printing using the UV-Vis polymerization of curable values [125]. On the other hand, Lopez-Dolado and team explored the potential of carbon nanoparticles coupled with 3D printing techniques. They used visible polymerization to create multi-layer 3D composite scaffolds, which exhibited excellent biocompatibility with the normal development of neural stem cells [126] (Figure 6d).

**Figure 6 ijms-24-05182-f006:**
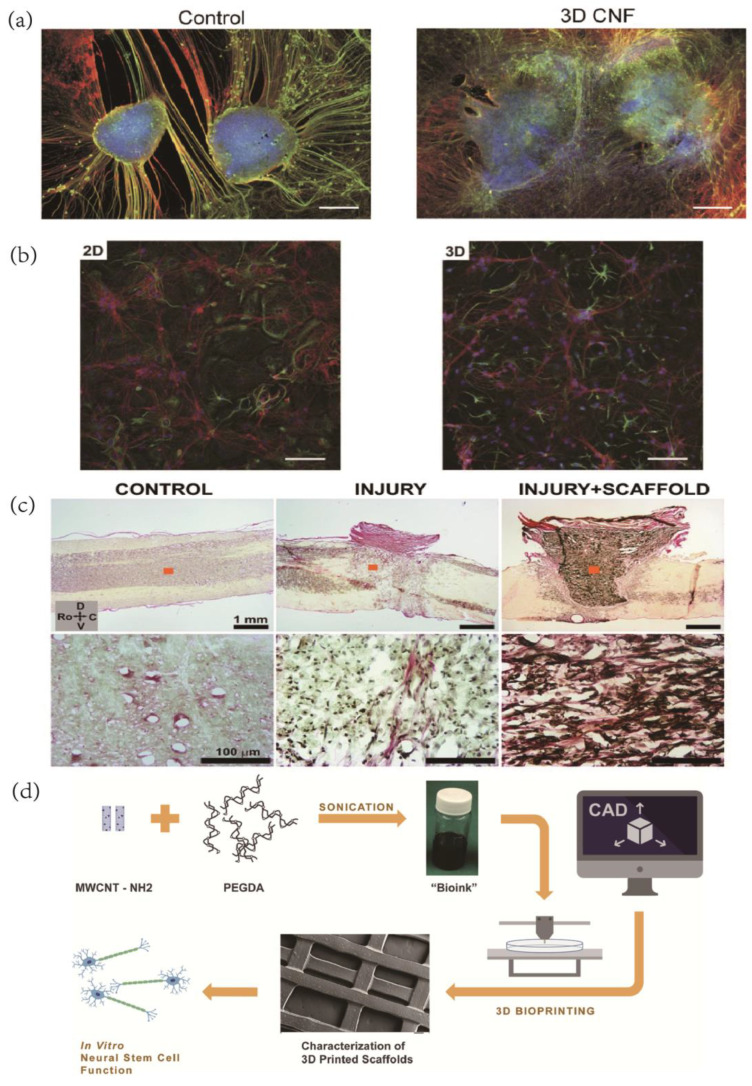
(**a**) Spinal organotypic slices cocultured in control and 3D CNTs after 14 days of growth. Immunofluorescence is for neuron-specific microtubules (b-tubulin III; red), neurofilament H (SMI-32; green), and nuclei (DAPI; blue). Reprinted with permission from [124] Copyright © 2023, The Authors. (**b**) Confocal micrographs show that hippocampal cultures grew. Reprinted with permission from [123] Copyright © 2023, Springer Nature. (**c**) Histological examination of the implantation site at 10 POD by HvG staining. Images in the bottom row represent zoom-in details of areas marked with orange squares in top images. Spinal cords are oriented in all cases as indicated by the set of arrows: C—Caudal, D—Dorsal, Ro—Rostral, and V—Ventral. Reprinted with permission from [126] Copyright © 2023, Wiley. (**d**) Schematic illustration of the overall experimental design. Reprinted with permission from [127] Copyright © 2023 IOP Publishing Ltd.

## 4. Concluding Remarks

Safety is a critical aspect of biomaterial research, particularly in the context of implantable BCIs. With the advancement of carbon nanomaterials, their biosafety has been a focus of research and discussion. Several studies have demonstrated the harmful effects of carbon nanotube exposure on cells, with metal impurities such as Fe, Ni, Co, Mn, and Y nanoparticles being the primary culprit. These metallic contaminants, which can breach cell membranes and produce reactive oxygen species, are often produced during mass production chemical vapor deposition processes, where catalyst residue contamination is unavoidable. However, the careful removal of metallic impurities and chemical functionalization of nanomaterials can significantly reduce their toxicity. Multi-walled carbon nanotubes have been shown to dynamically regulate synapse formation and function and modify sensitive neurobiological pathways, but may abolish cholesterol release regulation. Furthermore, the material’s structure significantly affects cellular exposure and extracellular toxicity. For example, graphene oxide and carboxylated graphene oxide mainly collected on the cell membrane, while hydrophilic carboxylated graphene oxide was internalized by the cells and accumulated in the perinuclear region without affecting cytoskeletal morphology. Nonetheless, research has demonstrated that graphene substrate materials can retain the properties of neuronal signals and are biocompatible with brain cell cultures.

In the realm of biocompatibility, the cytotoxicity of graphene and its derivatives have been a subject of significant study. While pristine graphene has been shown to promote programmed cell death, reduced graphene oxide is less cytotoxic. Notably, in tests to determine hemocompatibility, graphene does not cause erythrocyte hemolysis or affect coagulation pathways, indicating a low risk of intravascular thrombus formation. It is important to note that the toxicity of graphene electrodes is due to the electrode linker rather than the graphene material itself. Studies by Fabbro et al. demonstrate that graphene-based interfaces do not negatively impact target neuronal cells. However, while ideal graphene flakes do not enter lipid bilayers at room temperature, sharp protrusions or edge corners formed during production can infiltrate cells and cause irreversible damage. MXene materials have also been observed to induce cell membrane damage, acting as a “nano-thermal blade.”

Carbon nanomaterials have emerged as a highly valuable research tool for neuroscience, with a crucial role to play in the development of BCI (BCIs). These nanomaterials have been utilized in implanted BCI systems to record brain impulses and activate neural tissue cells, offering superior performance when compared to conventional metal and silicon electrodes by over an order of magnitude. While metal electrodes can be used in flexible structures for applications such as mesh structures, they come with biological toxicity and manufacturing cost constraints, as well as limited flexibility and high production costs. Carbon nanomaterials, on the other hand, offer flexibility and lower manufacturing costs, enabling them to be modified for drug transport and biosensing functions. These functional applications are typically not feasible with traditional electrode materials due to their lower flexibility and higher manufacturing costs. This research study provides a comprehensive overview of the latest advancements in implantable BCIs that incorporate various carbon nanomaterials.

The development of future BCIs will require the utilization of nanomaterial-based technologies to meet current objectives of interfacing with neurons. This includes improving the interface’s stability, flexibility, and persistence, as well as increasing the effectiveness of charge transfer to the neuron while minimizing the surrounding tissue’s response. The ultimate aim is to create highly precise, minimally invasive interfaces capable of detecting and mapping brain activity and providing targeted stimulation, as well as interfacing with external devices or facilitating functional restoration through engagement of neuroplastic processes that link the brain, spinal cord, and motor functions following injury [128]. Carbon nanomaterials are a promising solution for achieving electrode downsizing, neuronal scale stimulation, and recording. This would ultimately enhance the connection between electrical and biological systems [129]. With their unique properties, carbon nanomaterials have the potential to be an effective tool for developing high-performance BCIs, improving recording and stimulation efficiency, and advancing our understanding of neural functions.

Research on carbon nanomaterials has led to the development of a novel class of detection strategies, offering unparalleled precision and sensitivity for manipulating biological systems. This has had a significant impact on the basic neurobiology research for the application of carbon nanomaterials at the interface of implantable BCIs. Despite their potential as replacements for standard metal materials in BCI interfacing, carbon nanomaterials face several challenges.

Firstly, their production process is relatively complex, requiring high manufacturing and processing procedures that increase costs. Secondly, while carbon-based nanomaterials exhibit good electrical conductivity and have a large surface area, they cannot be effectively bonded to electrodes alone, leading to minimal improvements in electrochemical properties. Instead, they frequently form an interface modification layer with metals or conducting polymers, which enhances adhesion between the modified layer and the electrode.

Finally, many clinical applications require implants that function over several years, necessitating the urgent evaluation of carbon nanomaterials’ stability and safety over long timescales. Addressing these challenges is critical to realizing the potential of carbon nanomaterials in BCIs, including their ability to improve electrode downsizing, neuronal scale stimulation, and recording efficiency. By overcoming these obstacles, carbon nanomaterials may provide the foundation for the development of next-generation, high-performance BCIs with enhanced functionality and improved longevity.

Despite concerns regarding their in vivo behavior, the shift in medicine towards using multifunctional carbon nanostructures presents tremendous opportunities. These materials hold significant promise as a material application basis for advancing fundamental research in implantable electrode interfaces. With their unique properties, such as the ability to facilitate neural electrical signal stimulation and acquisition, support tissue engineering, and enable neuro bio-sensing, carbon nanomaterials can usher in a new era of innovation in the field of neuroscience. Such research has the potential to greatly enhance our understanding of neural processes and could lead to the development of novel medical devices and therapies for treating neurological disorders.

## Figures and Tables

**Figure 1 ijms-24-05182-f001:**
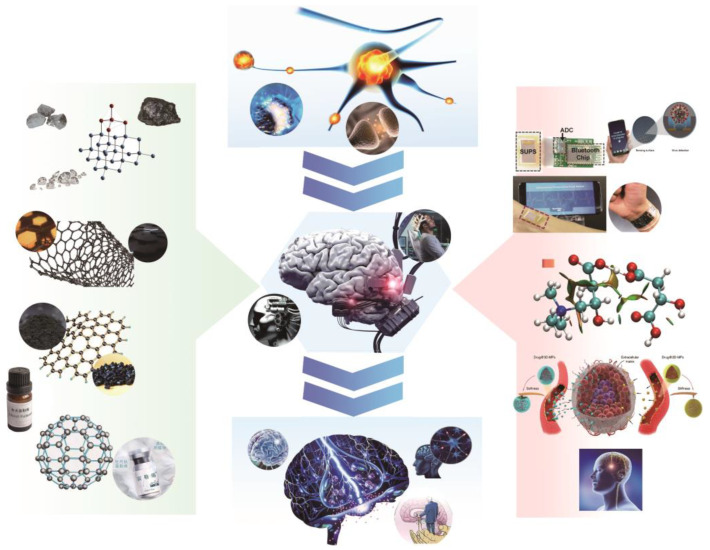
A brief but comprehensive survey of the utilization of carbon nanomaterials in implantable BCI. The central portion illustrates the translation of neuroscience mechanism research into BCI technologies for treating neurological disorders. The leftmost section outlines the different carbon nanomaterial structures that aid in BCI research, while the rightmost section presents various techniques for applying carbon nanomaterials in BCI, including neural interfaces, drug delivery, and biochemical sensing, etc.

## Data Availability

Not applicable.

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
