# Peer review of "Sensing and Stimulation Applications of Carbon Nanomaterials in Implantable Brain-Computer Interface"

_ijms, 2023, doi:10.3390/ijms24065182_

Round 1

Reviewer 1 Report

I commend the authors for providing a through review of the literature on the subject matter. My main issue is that the use of the English language is somewhat deficient and makes reading very difficult. There are many instances to name all here, but some are listed below:

- Title: add space between in and implantable

- Abstract: delete "interface" in line 17

- Introduction, line 29: replace "entering the 21st century" with "the beginning of the 21st century"

- Introduction, line 33: replace "has launched" with "launching"

- Introduction, line 35: delete "on the other hand"

- Introduction, line 36: replace "studies" with "programs"

- Introduction, line 46: define acronym BCI.

- Introduction, line 52: sentence in the middle of the line is incomplete, so it does not make full sense.

- Introduction, line 67: The sentence that starts "1985 at Rice University..." will be clear if restated as "The discovery of fullerene, the first member of carbon-based nanomaterials, by Smalley, Kroto, Curl and coworkers (1985) started a...."

 - Introduction, lines 72-73: The sentence that starts in line 72 and continues into line 73 is incomplete and therefore unclear.

-  Introduction, line range 83-92: This is a very long sentence with many descriptions which can be split into various sentences to make the message clearer.

- Line 132: Delete "Nikolas A". When referring to literature citation, use only the last name (surname) of the first author. There is not need to write the first name and middle initial. 

- Line 181: Delete "massless Dirac fermions" within the parenthesis as this is a repeat.

- Line 204: Delete "2008 EDWARD W KIEFFER" and replace with "Keefer"

- Line 206: Define the acronym MEA.

And many more. I recommend to have the manuscript proofread by someone well-versed in the language. 

Figure 1: The figure seems to miss some labels. What does it try to convey to the readers? Left column (green) seems to involve classes of carbon-based nanomaterials; right column (pink) seems to involve wearables and other information with very small text hard to read. The central column (blue) seems to indicate that advancements in the content of the left and right columns provide some implantable BCIs. Anyway, authors should improve labels or caption to help readers comprehend their message with this Figure. Also, make sure to acknowledge copyrights when appropriate.

Reviewer 2 Report

The manuscript by Jinning Li et al reviews the use of carbon-based nanomaterials for electrophysiological chemical sensing, tissue repair, and drug delivery. It primarily focuses on graphene and carbon nanotubes. The manuscript is well-structured and covers the most relevant works in the research field.

Although the review is informative to the scientific community, it could benefit from a more in-depth analysis of the limitations and difficulties of using carbon nanomaterials in the Conclusion section. This could include a brief comparison with other competing technologies to provide a more comprehensive understanding of the strengths and weaknesses of using carbon-based nanomaterials.

Overall, I believe that expanding the concluding section to address the limitations and difficulties of using carbon nanomaterials will make the manuscript more valuable to readers, as it will provide a more nuanced view of the potential of these materials.

Round 2

Reviewer 1 Report

Authors have addressed my concerns satisfactorily.